# Artificial-Intelligence-Driven Algorithms for Predicting Response to Corticosteroid Treatment in Patients with Post-Acute COVID-19

**DOI:** 10.3390/diagnostics13101755

**Published:** 2023-05-16

**Authors:** Vojtech Myska, Samuel Genzor, Anzhelika Mezina, Radim Burget, Jan Mizera, Michal Stybnar, Martin Kolarik, Milan Sova, Malay Kishore Dutta

**Affiliations:** 1Department of Telecommunications, Faculty of Electrical Engineering and Communications, Brno University of Technology, Technicka 12, 616 00 Brno, Czech Republic; 2Department of Respiratory Medicine, University Hospital Olomouc and Faculty of Medicine and Dentistry, Palacky University Olomouc, I. P. Pavlova 6, 779 00 Olomouc, Czech Republic; samuel.genzor@fnol.cz (S.G.);; 3Czech National e-Health Center, University Hospital Olomouc, I. P. Pavlova 6, 779 00 Olomouc, Czech Republic; 4Department of Respiratory Diseases and Tuberculosis, University Hospital Brno and Faculty of Medicine and Dentistry, Masaryk University Brno, Jihlavska 340/20, 625 00 Brno, Czech Republic; 5Centre for Advanced Studies, Dr. A. P. J. Abdul Kalam Technical University, Jankipuram Vistar, Lucknow 226021, India

**Keywords:** personalised medication recommendation algorithms, artificial intelligence, post-COVID syndrome, prediction model, respiratory system, corticosteroids, eHealth

## Abstract

Pulmonary fibrosis is one of the most severe long-term consequences of COVID-19. Corticosteroid treatment increases the chances of recovery; unfortunately, it can also have side effects. Therefore, we aimed to develop prediction models for a personalized selection of patients benefiting from corticotherapy. The experiment utilized various algorithms, including Logistic Regression, *k*-NN, Decision Tree, XGBoost, Random Forest, SVM, MLP, AdaBoost, and LGBM. In addition easily human-interpretable model is presented. All algorithms were trained on a dataset consisting of a total of 281 patients. Every patient conducted an examination at the start and three months after the post-COVID treatment. The examination comprised a physical examination, blood tests, functional lung tests, and an assessment of health state based on X-ray and HRCT. The Decision tree algorithm achieved balanced accuracy (BA) of 73.52%, ROC-AUC of 74.69%, and 71.70% F1 score. Other algorithms achieving high accuracy included Random Forest (BA 70.00%, ROC-AUC 70.62%, 67.92% F1 score) and AdaBoost (BA 70.37%, ROC-AUC 63.58%, 70.18% F1 score). The experiments prove that information obtained during the initiation of the post-COVID-19 treatment can be used to predict whether the patient will benefit from corticotherapy. The presented predictive models can be used by clinicians to make personalized treatment decisions.

## 1. Introduction

Severe acute respiratory syndrome coronavirus (SARS-CoV-2) is a pathogen responsible for coronavirus disease 2019 (COVID-19). This respiratory illness is responsible for significantly increased morbidity and mortality globally. Although most patients have only a mild course of the disease without any further complications, in more severe cases, pneumonia, acute respiratory distress syndrome, or even multi-organ failure may develop [1,2]. The virus usually enters the body via the respiratory epithelium (nasal cavity or lower airway). In its acute phase, it often replicates down the respiratory tract, and in some cases, it also causes significant lung damage, leading to pneumonia. These cases may have moderate or severe clinical courses. The progress of the disease is depicted in Figure 1. If the virus attack also hits the respiratory tract, it often leads to pneumonia or acute respiratory distress syndrome (ARDS). In addition, ARDS can cause many different long-term sequelae, many of which can also be severe.

The post-acute phase is commonly considered a period after 3 or 4 weeks after the acute phase [3,4]. Post-acute syndrome is a disease that affects different body systems and has complex long-term effects on patients [5]. In more severe cases, persistent pulmonary interstitial damage may develop. The most typical finding is organizing pneumonia, which sometimes may progress to irreversible fibrotic changes. The treatment of persistent lung interstitial damage in post-acute COVID-19 is still not standardized. For example, according to the Czech national position document for the treatment of pulmonary involvement in COVID-19 [6], corticosteroid (CS) treatment is indicated in cases of prolonged resorption of pulmonary infiltrates as a prevention of the development of fibrosis as a long-term consequence. Similarly, Bieksiene et al., in their systematic review, state that appropriately timed CS therapy may be beneficial for selected patients [7].

This paper focuses on treating interstitial lung damage in the post-acute phase of COVID-19. The treatment was indicated in concordance with the Czech national position document [6]. Only patients with persistent lung interstitial involvement (all of them had COVID-19 pneumonia in the acute phase) were indicated for CS treatment. This study aims to develop AI-based algorithms that help correctly distinguish between post-COVID patients for whom CS therapy is recommended (i.e., they will not spontaneously recover, and CS can help them) and those who recover without CS therapy. Thus, the developed algorithms can prevent overtreatment.

The rest of the paper is structured as follows. Section 2 describes related work, which discusses possible post-acute COVID-19 complications and focuses on mortality, complications like pulmonary fibrosis, and predictive machine learning approaches. It also identifies several areas that are still waiting for evidence-based studies. Section 3 describes the data used for the experiment and applied machine learning methods. Section 4 presents the results of the experiment and discusses their meanings. Section 4.2 describes the limitations of this study and possible further possible directions. Section 5 concludes the paper.

## 2. Related Work

Most people with COVID-19 have mild to moderate symptoms. However, some of them may develop a systemic inflammatory response of the body, resulting in lung damage, multisystem organ dysfunction, or long-term health complications reducing the quality of life as a result of the disease. These severe consequences can be prevented or alleviated by using anti-inflammatory drugs, such as CS. However, despite the presence of official CS treatment processes that national institutions have approved [6,8], there is no absolute consensus among a wide range of physicians and specialists on the target patient’s group, benefits, and scheduling of CS administration, as well as duration and dosage. Due to the mentioned facts with the combination of the global spread of the SARS-CoV-2 virus, many studies are being conducted to answer raised questions related to CS therapy in the acute and post-acute phases of COVID-19. However, many published research on the efficiency and benefits of CS in treating patients with severe consequences contradict one another.

### 2.1. Acute-Phase and Corticotherapy

#### 2.1.1. Administration Timing and Dosing

Fadel et al. [9] focused on the impact of CS administered early in patients with moderate to severe course. A total of 213 patients were included, with 132 receiving CS (0.5–1.0 mg/kg/day of Methylprednisolone divided into two doses); the rest received standard care. The median duration of hospitalization was significantly reduced in the CS group—from 8 to 5 days. Another study [10] supports the conclusions of the previous paper and did not find significant differences in mortality between patients receiving high doses of CS and those who did not. The trial included 422 patients, more than half of them receiving CS treatment (Methylprednisolone or Tocilizumab). However, both studies focus on the acute phase and do not study long-term consequences.

The authors [11] discuss the advantages of using low doses of CS to treat patients with COVID-19 pneumonia when administered on time. The purpose of the study is to find whether late high-dose CS therapy has a similar effect. A total of 348 patients participated in the analysis. They received a median dose (1 mg/kg) of Methylprednisolone equivalent. The treatment started 21 (18–26) days after the onset of symptoms. They concluded that late high-dose CS had no benefit in reducing intensive care unit (ICU) mortality nor in suppressing later acute respiratory distress syndrome development.

According to Monreal et al. [12] positive benefits of high-dose corticosteroid treatment were not observed even when administered early. In the study, 573 patients were included. The majority of patients were men −74.70% with a median age of 64 years. Standard doses (1.0–1.5 mg/kg/day) were administered to 379 patients (69.10%) and the rest received high doses (250–1000 mg/kg/day). In high-dose treatment group were observed even higher mortality and an increased risk of mechanical ventilation. This phenomenon has been observed mainly in the elderly. It should be noted that the paper reports bias in the experiment, the group of patients who were prescribed a higher dose consisted of elderly patients who suffered much more often from some comorbidity and at the same time showed worse respiratory function.

Similar results are presented in an observational study of 1379 individuals [13]. In total, 873 of them received more than 40 mg of Methylprednisolone equivalent; the others received a lower dose. A higher dose of CS was not associated with a change in the inflammatory markers or duration of mechanical ventilation. This group of patients had a higher infection rate but a decreased risk of acute kidney injury. The authors also conclude that combining Tocilizumab and CS was proven to be an effective therapy method. Compared to CS alone or standard therapy, this combination resulted in a higher survival rate [14]. A meta-analysis [15] involving 18,702 patients confirmed these conclusions as well as the efficacy of the combination of Tocilizumab and CS. Previous studies show that CS treatment is beneficial when used early with a recommended dose of 1.0–1.5 mg/kg/day. However, later treatment with a higher dose sometimes produces the desired benefits, and the patient’s health can worsen. Treatment outcomes also improved when CS and Tocilizumab were used together. They conclude that late high-dose CS had no benefit in reducing ICU mortality nor in suppressing later acute respiratory distress syndrome development. Another study [10] found no significant differences in mortality between patients receiving high doses of CS and those who did not. The trial included 422 patients, with more than half receiving CS treatment (Methylprednisolone or Tocilizumab).

#### 2.1.2. Mortality

Some publications, rather than dosing and timing, focus on the mortality rate. The main objective of study [16] was to discover how CS therapy affects 28-day mortality. The patients were divided into two groups; the first received Dexamethasone at a dose of 6 mg per day for ten days, while the second received standard treatment. The trial analysed a total of 6425 patients, 2104 of whom were administrated CS. The finding indicates that when patients required respiratory support (mechanical lung ventilation of oxygen therapy), the mortality rate in the CS group was lower than in the standard treatment group (29.3% vs. 41.4% and 23.3% vs. 26.2% respectively). Another significant finding is that the mortality ratio was higher in the CS group patients who did not require respiratory support (17.8% vs. 14.0%). The authors of the study [17] present a distinct view on the benefits of CS therapy. A comprehensive meta-analysis of 21,350 patients concluded that the overall 28-day mortality rate was higher in those treated with CS for 3 to 12 days (also observed in a group of patients receiving high doses of CS [13]). As a possible explanation, they mention the prothrombotic influence of CS in combination with the response to other administered drugs. They recommend utilizing CS after carefully considering the benefit-to-risk ratio for a maximum of ten days of therapy.

Despite conflicting findings on CS therapy’s benefits, their effect is generally perceived positively. There are also some healthcare guidelines [6,8] for the treatment of the acute phase of the disease. However, due to the relatively strong side effects, CS should not be administered to everyone; instead, only those who will benefit from the treatment should be identified. Therefore, the official procedures also define certain situations in which CS is recommended. Current research is focused on confirming existing processes or finding new factors that could be used to identify patients who will benefit from CS.

#### 2.1.3. Machine Learning Approaches

Machine learning has great potential in personalizing the treatment and application of CS more specifically. It was already used, for example, in [18], where a machine learning model was designed to estimate the level of in-hospital mortality in patients treated with CS. A total of 1571 participants were included in the analysis. The proposed light gradient boost model has a high level of accuracy (AUC—0.881). Work [19] proposes a machine learning-based approach, which identifies patients for whom treatment with CS or Remdesivir will increase survival time. The method is based on the Gradient-boosted decision-tree model. The dataset used to train the model includes health status from 2364 patients acquired from 10 US hospitals. The authors describe some limitations of the work, including retrospective character: it is unknown how treatment recommendations can impact prescribing practices and patient outcomes in clinical settings. According to the authors, this work is the first one which applied the machine learning method to evaluate the effectiveness of treatment. There is also a work that utilized more machine learning algorithms [20]. The objective of the presented approach is to evaluate the response to CS therapy; the dataset consists of data from 666 patients. The used ML algorithms are Logistic Regression (LR), Support Vector Machine (SVM), Gradient Boosted Decision Tree (GBDT), *k*-Nearest Neighbor (*k*-NN), and Neural Network (NN). The results seem promising: AUC of 0.81 in the internal validation set and 0.85 in the external validation set. In addition, they tried to use the unsupervised machine learning approach (clustering) to evaluate response to CS therapy and the association between CS treatment and mortality [21]. The conclusion is that CS has a positive effect on survival in critically ill patients with the hyper-inflammatory phenotype. The [22] research presents a framework that involves artificial intelligence-based algorithms to predict the progression and prognosis of patients with COVID-19 based on the analysis of X-ray images supplemented with other patient data such as age and comorbidities present. The proposed method achieved 88% accuracy and 79% recall.

Unfortunately, the results obtained from the acute phase studies have limited impact on the long-COVID consequences.

### 2.2. Post-Acute Phase and Corticotherapy

Despite the successful recovery from the acute COVID-19 disease, people may continue to face long-term health issues. It happens to approximately one in three patients with a symptomatic course of COVID-19 disease. They suffer from at least one health sequelae even 12 weeks after infection [23,24]. Well-known long-term health problems include difficulty breathing, fatigue, cough, fever, depression, olfactory loss, muscle and chest pain, memory or sleep problems, etc. [25]. Additionally, one of the most serious health sequelae is pulmonary fibrosis because it can significantly reduce patient quality of life and decrease the length of life. One of the possible treatments for these long-term sequelae, similar to the acute phase of the disease, is CS therapy. For example, it has been successfully used in alleviating olfactory loss [26,27].

According to Myall et al. [24], approximately 39% of patients with COVID-19 pneumonia remain symptomatic following the post-acute phase of the disease. In their observational study, these patients were further examined using pulmonary function tests (PFT). Furthermore, a high resolution computed tomography (HRCT) examination was performed in patients with a substantial decrease in carbon monoxide (DLCO) diffusion lung capacity. CS treatment was indicated in 4.80% (35 subjects). The radiological result supported the pneumonia diagnosis, but histology verification was not conducted. All participants treated with CS had a substantial improvement in subjective dyspnea and a significant increase in DLCO (mean increase of 31.60%). The lack of a control group without CS therapy is a principal limitation of the study.

### 2.3. Summary and Research Gap

To summarise the overview, there is no absolute consensus on the benefits of CS treatment. Unfortunately, it is difficult to compare studies. Some studies examine the effect of the time of CS administration from the onset of symptoms, the optimal dose, the impact of treatment on patient mortality, hospital length of stay, days in the intensive care unit, and other factors. Studies often compare results in different periods, some of them in and just after the acute phase, some of them in +1 month, +3 months, or +6 months. The target group of patients treated by CS also often differs. CS is sometimes administered to elderly patients in critical condition and, at the same time to those who suffer from one or more comorbidities. The authors mention it as a possible influence on the study outcomes. Nevertheless, some studies report positive effects [16,24], but many studies claim the opposite [17]. Larger unified datasets can help to overcome those problems in the future.

Another limitation is the level of detail that the studies use about each patient. Most of them use just a basic information set (age, sex, height, sometimes selected symptoms, etc.). However, more information is commonly available at the start of post-acute COVID-19 treatment (e.g., blood tests, functional lung tests, symptoms, patient status, and lung X-ray or HRCT), and it is a pity that these data are not shared between researchers. In addition, today’s size of datasets does not clearly state when the treatment should be administered and when not; also, personalization of the treatment administration could be a benefit to the patients.

There are still many open questions regarding CS treatment. Especially in which benefits outweigh their side effects (i.e., when they should be applied), when they should be administered, their optimal dosage, and how long the treatment should last.

With a high probability, the quality of life will also be reflected in the increased morbidity in the future. However, this is expected to be reflected in the long-term horizon (5+ years). Unfortunately, there is no study devoted to post-acute COVID-19 CS treatment. There are only those that focus on the issue of the acute phase of the disease. Artificial intelligence-based approaches have the potential to increase accuracy and can be personalized. Currently, there has yet to be any work devoted to this topic so far. The contribution of this paper is artificial intelligence-based algorithms for personalizing CS treatment in patients with a risk of developing pulmonary fibrosis due to COVID-19.

## 3. Methodology

This section describes the methodology used in this experiment. First it describes a dataset, in particular, who patients were enrolled in the study, and which parameters were examined during the initial examination (see Section 3.1). It also describes, which machine learning algorithms and optimization techniques were used (see Section 3.2). The metrics used for the evaluation are described in Section 3.3. The source codes and the dataset are uploaded to the public repository (the link is in the Appendix A section).

### 3.1. Dataset

#### 3.1.1. Patient Selection Process

This experiment enrolled a total of 1861 patients, who had confirmed COVID-19 disease. The patients who have not been affected by respiratory track with pneumonia were excluded. The reason is, that occurrence of PF in patients with no pneumonia is quite rare, so CS treatment is not considered in those cases. Moreover, we examined possible signs of pre-existing interstitial lung disease—either by the typical picture on HRCT scans or by biopsy in uncertain cases. Only subjects with clear post-COVID lung damage were involved in the analysis. In total 1580 patients were excluded so only those patients for whom CS is considered are enrolled in the study. The algorithm of selection is depicted in Figure 2.

#### 3.1.2. Analysed Dataset

The dataset used for the machine learning algorithm analysis enrolls 281 patients, where 60.5% were males and 39.5% were females (see Table 1 for detailed demographics information). The objective of the study is to predict, whether a patient will suffer from PF in 3 months horizon or whether the complications will regress, and thus whether the CS treatment is recommended or not.

The patients are split into two groups, (1) patients who received CS therapy as post-COVID treatment (95) and (2) those who did not (186). The patients who received CS therapy were those who suffered from persistent pulmonary interstitial damage induced by COVID-19 pneumonia. The received dose was 0.5 mg of prednisolone per kilogram of body weight (with a maximum dose of 40 mg) for 2 weeks, with subsequent 4 weeks of 20 mg of prednisolone treatment and further gradual tappering of the doses until withdrawal. It should be noted that CS could not be administered to patients with bacterial or other infections. The exact details are included in the Czech national position document [6].

Each patient was subjected to an initial examination at the start of post-COVID treatment. It is approximately 3 weeks after the first visible symptoms of the acute COVID-19 disease (see Figure 1). The data collected for the experiment include physical examination (age, body mass index, is a smoker, presence of other comorbidities, etc.), information related to acute-phase treatment, pulmonary function tests data, and blood tests data (e.g., immunoglobulins IgG, IgM).

The data also include information about the results of the treatment after approx. 3 months after the start of the post-acute treatment, i.e., whether the patient is getting better or not. This information is represented by an objective rating based on results of X-ray, CT and a subjective rating based on feedback from the patient.

During the selection of the treatment, we tried to minimize possible bias as much as possible. To reduce the bias and the risk of overfitting, some attributes from the dataset were removed. The possible bias can occur because of not a random selection of the patients, whether they received CS or not. The complete dataset was released, including detailed information about each attribute to make it easier to integrate with other datasets in the future. In this paper, only those parameters, which were identified as significant, are explained. Demographic information about the dataset is shown in Table 1. The data from spirometry contains for each attribute values marked “(pred)”, “(abs)” and “(%pred)”, where “(pred)” stands for predicted normal value based on height, age, etc., “(abs)” stand for the actual absolute measured value. For the analysis only the “(%pred)” and “(abs)” values were used, since they reflect the expected body composition of each patient according to their height, body mass index etc.

From the point of the statistical analysis, it would be optimal to select the patients for CS treatment on a random basis. Since the main priority is patients’ health and providing them with the best possible treatment, this was not possible and would be considered unethical.

An overview of selected qualitative parameters is shown in Table 1. It contains information about both groups of patients: (1) who received and (2) who did not receive CS treatment. Percentages from the total number of respective groups for each parameter are also provided.

For each patient, the objective radiological score for lung damage regression is defined in the range of 0 (immutable state) to 10 (complete regression); these cases can be seen in Figure 3. The score is a decisive parameter in determining whether the patient’s health has improved. Patients with a score of less than seven are assessed as not significantly improving their health status. Based on this information, the patients are divided into two groups, one of which is recommended CS (n=135) and the other not (n=146).

The entire dataset was divided into training and testing subsets in a ratio of 80 to 20%. The training set contains 224 patients; 116 are not recommended to be administered with CS, while others are recommended to be treated with CS (n=108). The testing set contains 57 patients divided into two groups, one of whom is recommended corticotherapy (n=27) and the other not (n=30).

### 3.2. Experiment

In this experiment, we examined nine machine learning algorithms, which aim to perform classification task and predict whether the CS treatment should be recommended for a patient or not.

#### 3.2.1. Feature Selection

The first step is to select the set of features that will be used for algorithms. For this purpose, we used information from SHapley Additive exPlanations (SHAP) [28], which evaluated the impact of features on the output of some algorithm, in our case—Decision Tree, this output is shown in Figure 4. One of the most important features seems to be the amount of CS received during the treatment and IgM values from blood tests. We also used the method of selecting *k*-highest scores based on *p*-values.

After that, some features were selected from the output of both methods. The common features from both methods are the amount of used CS, SARS-CoV-2 IgM(quant.), and Blood test Mo %. With the *k*-best method, we selected the following features: corticosteroids use, duration of CS use in weeks, total used CS, olfactory loss, blood test RDW, VC(abs), FEV1(abs), and PEF(% pred). Additionally, some other features, which have an impact on the decision of treatment from the medical side, were added: pneumonia, comorbidities, post-COVID disability, SARS-CoV-2 IgG(qualit.), FVC(% pred), DLCOc SB(abs), KCO SB(abs), MEF25(abs), persistent cough and persistent dyspnea. The feature selection is depicted in Figure 5.

The final set of features contains the following: pneumonia, comorbidities, corticosteroids use, olfactory loss, post-COVID disability, SARS-CoV-2 IgG(qualit.), SARS-CoV-2 IgM(quant.), amount of used CS, total used CS, duration of CS use in weeks, blood test RDW, VC(abs), FVC(% pred), FEV1(abs), Blood test Mo %, PEF(% pred), DLCOc SB(abs), KCO SB(abs), persistent cough, persistent dyspnea and MEF25(abs).

#### 3.2.2. Selected Methods

The selection of the algorithms used in the experiment was based primarily on their ability to analyse smaller datasets and their potential for easy explanation, which is particularly relevant to the Decision tree algorithm. The selected algorithms are listed below:Logistic Regression [29] (pp. 89–90),*k*-Nearest Neighbours [29] (pp. 56–59),Decision Tree [29] (pp. 167–169),XGBoost [29] (pp. 190–193),Random Forest [29] (pp. 194–195),Support Vector Machine [29] (pp. 145–146),Multi-layer Perceptron (MLP) [30],Adaboost classifier [29] (pp. 190–191),Light Gradient Boosting Machine (LGBM) [31].

The training was performed with a combination of preselected hyperparameters using a random search of 150 iterations and partially manually optimized. The models were evaluated using *k*-fold cross-validation with k=5. This step is used to find the best parameters for higher accuracy, and the training process is performed on the training set.

The optimal parameters, which were found used with the respective models are:MLP: hidden layer sizes: (70, 8), optimizer: Adam, max iteration: 80, activation: ReLU;Decision tree: min samples leaf: 13, max depth: 8, criterion: entropy;Random forest: max depth: 6, criterion: entropy;*k*-Nearest Neighbours: weights: distance, neighbors number: 5;SVM: kernel: sigmoid;AdaBoost: number of estimators: 12, learning rate: 0.8;LGBM: learning rate: 0.5, max depth: 3.

### 3.3. Evaluation Metrics

We selected several metrics to evaluate and compare the results from the experiments described in the previous section. These metrics include accuracy (see Equation (Equation 1)), balanced accuracy (see Equation (Equation 5)), $F_{1}$ score (see Equation (Equation 2)), sensitivity (see Equation (Equation 3)), and specificity (see Equation (Equation 4)). For a description of the following metrics, there were used the following abbreviations: TP—number of true positive cases, TN—number of true negative cases, TN—number of true negative, and FN—number of false negative cases.

Accuracy measures the ratio of correctly predicted labels over total number of evaluated samples [32]:(1)$$Acc=\frac{\mathrm{TN}+\mathrm{TP}}{\mathrm{TP}+\mathrm{TN}+\mathrm{FP}+\mathrm{FN}}.$$

$F_{1}$ score is a combination of precision and recall metrics, which capture properties of them both [32]:(2)$$F_{1}=\frac{\mathrm{TP}}{\mathrm{TP}+\frac{1}{2}\left(\mathrm{FP}+\mathrm{FN}\right)}.$$

Sensitivity measures, how the model can correctly predict positive samples [32]:(3)$$Sen=\frac{\mathrm{TP}}{\mathrm{TP}+\mathrm{FN}}.$$

Specificity measures, how the model can correctly predict negative samples [32]:(4)$$Spe=\frac{\mathrm{TN}}{\mathrm{TN}+\mathrm{FP}}.$$

Balanced Accuracy is average value between sensitivity and specificity [33]:(5)$${Acc}_{\mathrm{BAL}}=\frac{Spe+Sen}{2}.$$

## 4. Results and Discussion

This experiment aims to develop AI-based algorithms that would classify the necessity of CS therapy based on the initial examination of patients. The examination included the blood test, pulmonary test, and information about persistent health issues, including the assumption of treatment application and the possible amount and period of application.

In total, nine machine learning algorithms were evaluated on the dataset of 281 patients (224 were used for training and 57 for testing). The results of these experiments are shown in Table 2. As can be seen, the balanced accuracy is in the range of 61–73%. The best results were achieved by the Decision tree, 73.52%. The methods that reached 68–70% of balanced accuracy, are *k*-nearest neighbours (68.15%), multilayer perceptron (69.81%), random forest (70.00%), AdaBoost (70.37%). The classifiers with worse results are: XGBoost (61.11%), logistic regression (61.30%), support vector machine (61.85%), LGBM (62.59%).

### 4.1. Explainable Recommendation Approach

According to the results presented in Table 2, the best results achieved by the decision tree with balanced accuracy 73.69%, $F_{1}$ score—71.70%, precision—73.08%, ROC-AUC—74.69%. The main advantage of this method is that it can be easily graphically represented and is explainable to humans. From the practical side, such representation is a benefit for clinicians because it is possible to control the results of prediction and can be evaluated from the medical point of view. The resulting decision tree is shown in Figure 6.

From this decision tree, the first point which should be decided is: if the CS is going to be applied. This question is the assumption and the following leaves in the decision tree will help to answer the question if CS really should be applied.

In case it was considered to apply treatment, three other parameters should be checked: RDW and SARS-CoV-2 IgM from a blood test and persistent dyspnea. It is recommended to apply CS in two cases, and in the other two cases, it is not known. This answer appeared because there are the same number of patients who got well and did not get well, which is why it is impossible to say if the patient should get treatment or not.

On the other side, if it is selected not to give the corticosteroids, there can be different options. In some cases, it is recommended to apply corticosteroids, and in some cases not. If SARS-CoV-2 IgM is above 9.19, leaving the patient without CS is suggested. Otherwise, there can be different scenarios, which depend on FEV1(abs), KCO_SB(abs), MEF25(abs), and persistent cough. More detailed information is provided in Figure 6.

The confusion matrix is also presented in Table 3. 42 of 57 patients from the test set were predicted correctly: 23 cases, where it is not recommended to apply CS, and 19, where it is recommended. The 15 cases were wrongly classified.

Despite the good results of other methods, such as *k*-nearest neighbours, multilayer perceptron, random forest, and AdaBoost, which are also competitive with a decision tree, this method would be preferred in application in real practice.

### 4.2. Limitations

We admit that our dataset is only of modest size (281 included patients); therefore, we cannot train highly effective AI models only on our data. However, our study shows an example of such an AI together with real data—as a possible model example for other researchers—either for COVID-19 or other future viruses. The efficiency of 73.52% is still higher than the effectiveness of the clinician’s decision, as we are still missing any clear guidelines for CS medication in treating post-COVID lung involvement. We encourage the other teams to validate the proposed model with their data from their clinical practice. We also encourage other teams to share the data. Merged datasets can lead to more complex and reliable models.

The results obtained could be biased for a specific group of people. For example, race and ethnicity can influence the variance of the cohort. The presented data samples were obtained from the Olomouc region in the Czech Republic (Central Europe). In some former studies, it has been reported that different races can be more vulnerable to the COVID-19 disease [34], so different parameters could be of other importance in different regions. Validation of whether the findings of this study also fit other races is recommended. On the other hand, this study tried to prefer stable attributes and as independent of other factors as possible.

Furthermore, there might be possible selection bias in patients indicated to the examination by the clinicians as the majority of patients sent to the pulmonary department were symptomatic. Another limitation is that only a tiny proportion of the patients had a lung biopsy, and the treatment was based mainly on clinical status, radiology findings, and pulmonary function tests. Patients with known pre-existing interstitial lung disease were excluded from the analysis. However, there might be a theoretical chance of possible pre-existing interstitial lung disease in a small proportion of the patients, as in some patients, the treatment of COVID-19 infection was the first contact with the health care system. All the patients remain in our follow-up for at least the next two years to confirm the durable recovery from the lung damage.

It is also unclear what the responses of the human body of people with several comorbidities are. Moreover, the comorbidities could influence the physiological parameters and distort the patterns.

## 5. Conclusions

This paper presents the first AI-based algorithms for recommending subsequent medication for patients with pulmonary fibrosis, one of the most severe long-term consequences of COVID-19. These algorithms help identify patients with post-acute syndrome who will benefit from CS and who will not. The data used in this study includes clinical, functional, and imaging data about each patient. In total, 281 patients were enrolled in the study. Every patient suffered from COVID-19 pneumonia in their acute phase.

The proposed AI-based model reached 73.52% balanced accuracy. Some other models, such as *k*-nearest neighbors, multilayer perceptron, random forest, and AdaBoost, achieved 68.15%, 69.81%, 70.00%, 70.37% for balanced accuracy, respectively.

We also introduced a decision tree model for clinicians who prefer simple and interpretable models (see Figure 6). Clinicians also validated the model, and it was concluded that the principle also makes sense from clinical practice and is based on attributes that are accessible. The main contributions of this paper are:The artificial intelligence-based (or also called evidence-based) model predicts with 73.52% accuracy whether treating a patient with CS therapy is recommended. Pulmonary fibrosis is one of the most severe long-term consequences of COVID-19. We also identified the most valuable attributes used for the classification.Dataset FNOL_PulFib2022 (https://github.com/VojtechMyska/AI_CS_response, accessed on 12 May 2022) of 281 patients from post-acute treatment. Each patient is subjected to rigorous testing (e.g., blood tests, spirometry, anamnesis, and comorbidities) at the start of the post-COVID treatment and the result three months after the treatment. In addition, information on whether the patient benefited from CS treatment is also included.A simplified interpretable decision tree-based model, which can be easily incorporated into the clinical practice, is also provided.

Future work should focus on extending the experiment with more data, which should be the highest priority. Especially merging data from various sources can potentially reach exciting improvements. We encourage other teams to publish the data so the research community can benefit. Furthermore, experiments with imaging approaches, including X-ray or HRCT might lead to interesting results. Unfortunately, this would be data-demanding. Although this study also included comorbidities, their results cannot be considered statistically significant due to their rare occurrence. Hopefully, they will help in some follow-up studies. 

## Figures and Tables

**Figure 1 diagnostics-13-01755-f001:**
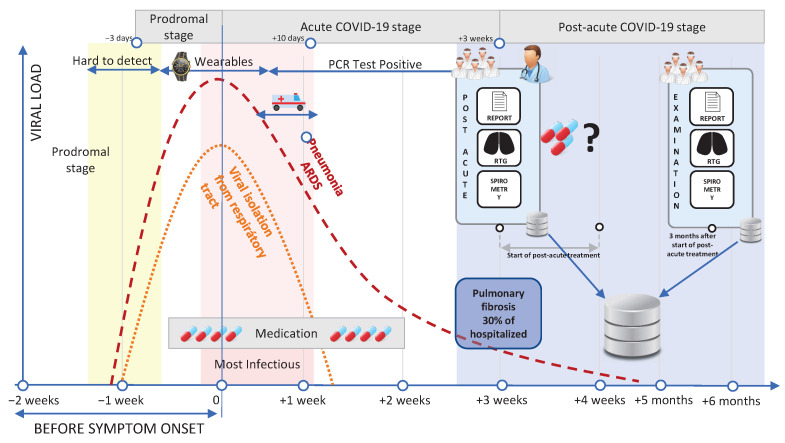
Process of the COVID-19 disease including onset, acute and post-acute stages of the disease.

**Figure 2 diagnostics-13-01755-f002:**
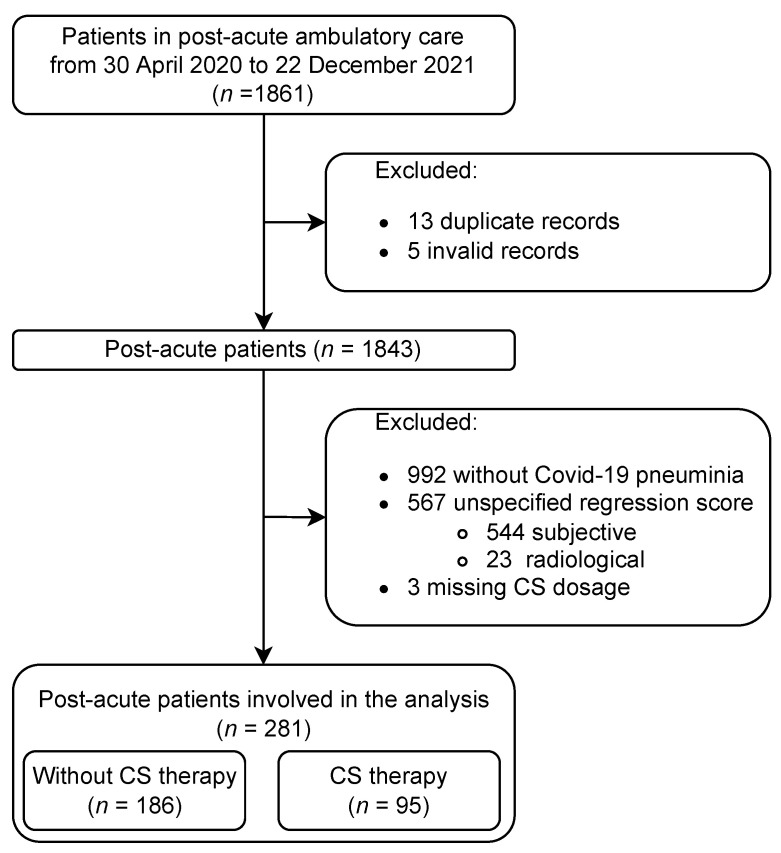
Inclusion of patients in the analysis with reason for exclusion.

**Figure 3 diagnostics-13-01755-f003:**
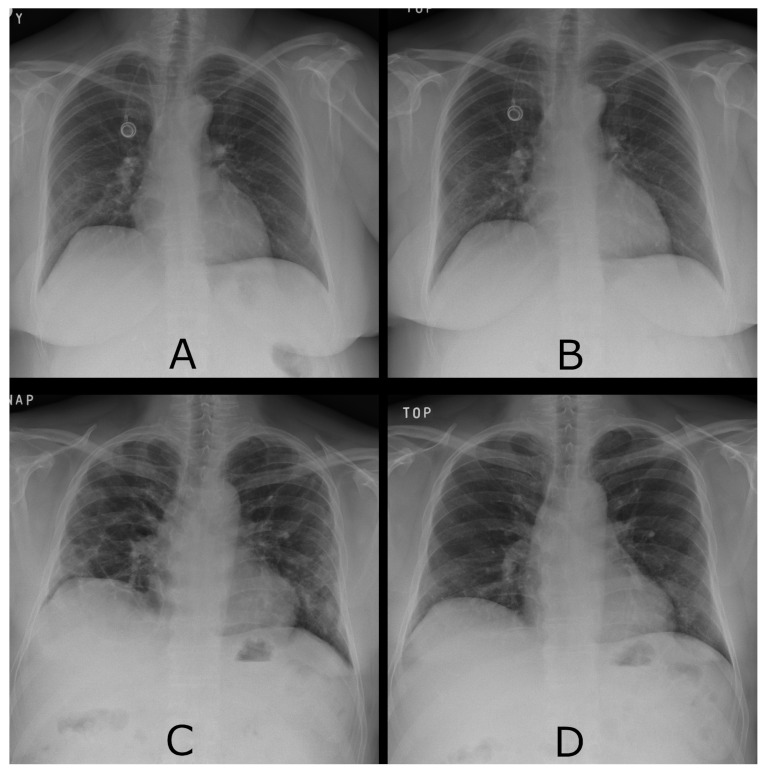
(**A**) The chest X-ray of a 49-year-old woman at the beginning of the acute phase of COVID-19 shows bilateral pulmonary infiltrates, dominantly on the periphery of both lungs. (**B**) The chest X-ray of patient A’s lungs after approximately three months. Minimal changes, regression score 0. (**C**) The chest X-ray of a 61-year-old man at the beginning of the acute phase of COVID-19. (**D**) X-ray of patient C’s lungs after approximately three months. Complete regression, regression score 10.

**Figure 4 diagnostics-13-01755-f004:**
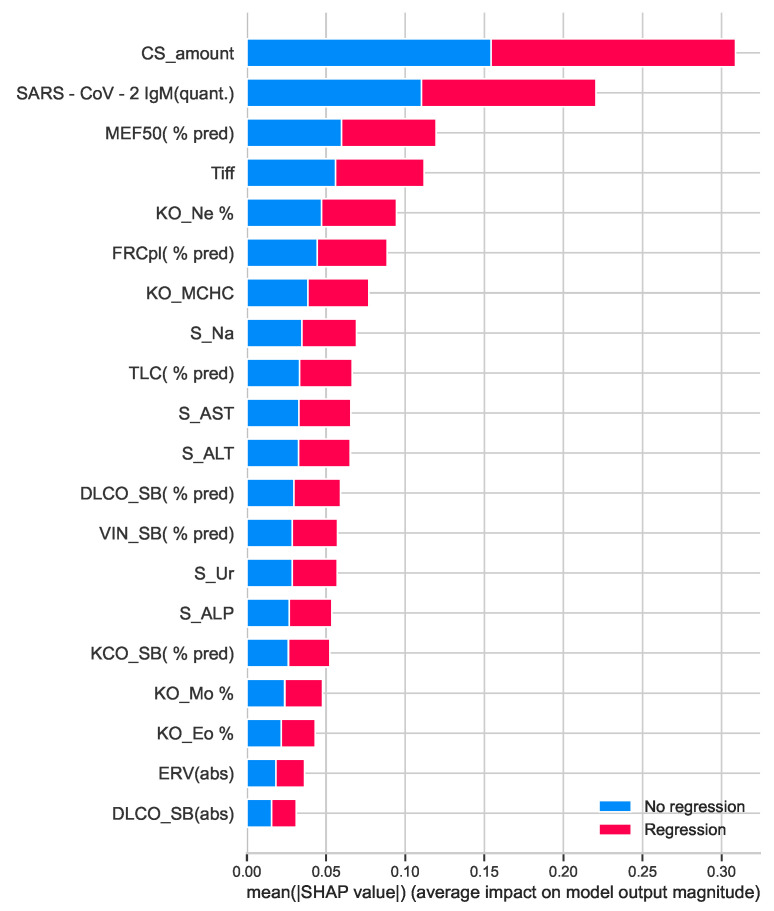
Feature importance based on SHAP analysis.

**Figure 5 diagnostics-13-01755-f005:**
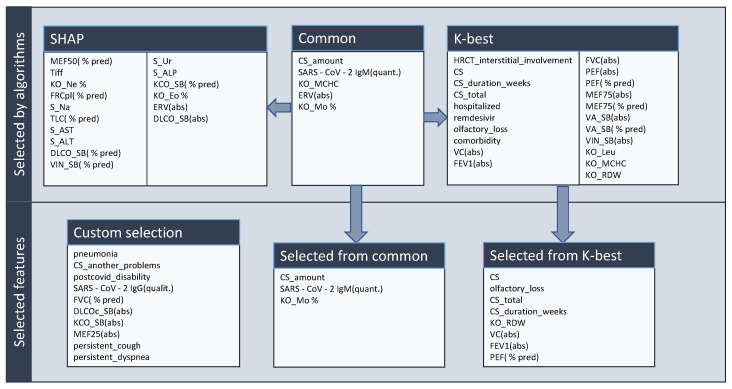
Selection of features for further machine learning.

**Figure 6 diagnostics-13-01755-f006:**
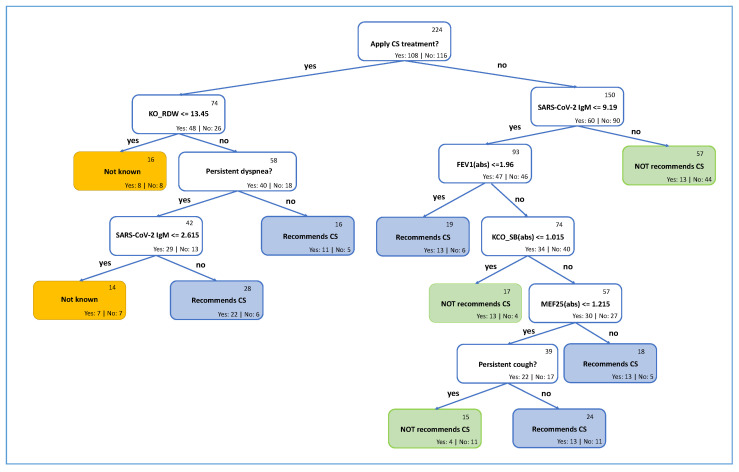
Decision tree model for recommendation of CS medication.

**Table 1 diagnostics-13-01755-t001:** Overview of demographics, habits and clinical data of patients included in the analysis.

DEMOGRAPHIC & HABITS					
**Attributes**	**Values**				
Number of patients	281				
**Gender**	**Male**		**Female**		
Number of patients	170 (60.50%)		111 (39.50%)		
**Age**	**Mean (SD)**	**Median (Q1–Q3)**		**Minumum–maximum**	
Years	64.33 (11.08)	65 (58–72)		30–90	
**Body proportion**	**Mean (SD)**	**Median (Q1–Q3)**		**Minumum–maximum**	
Weight (kg)	88.03 (15.71)	86 (77–97)		57–136	
Height (cm)	169.97 (9.76)	171 (163–176)		145–198	
BMI	30.49 (4.96)	29.71 (26.88–32.89)		20.75–47.37	
**Smoking**	**Smoker**	**Ex-smoker**	**Non-smoker**	**N/A**	
Number of patients	11 (3.91%)	55 (19.57%)	201 (71.53%)	14 (4.99%)	
**THERAPY & LUNG DAMAGE**					
**Attributes**	**Number of patients**				
	Yes		No		
Hospitalized	230 (81.85%)		51 (18.15%)		
Oxygen	185 (65.83%)		96 (34.17%)		
Remdesivir	22 (7.83%)		259 (92.17%)		
CS					
During hospitalization	102 (36.30%)		179 (63.70%)		
Post-covid treatment	95 (33.81%)		186 (66.19%)		
Another diagnostics	4 (1.42%)		277 (98.57%)		
HRCT—Lung damage					
Interstitial involvement	51 (18.15%)		230 (81.85%)		
Inflammatory changes	125 (44.45%)		156 (55.45%)		
**PERSISTENT HEALT ISSUES**					
**Attributes**	**Number of patients**				
	**Yes**		**No**	**N/A**	
Dyspnea	194 (69.04%)		86 (30.60%)	1 (0.36%)	
Cough	98 (34.86%)		182 (64.78%)	1 (0.36%)	
Fatigue	80 (28.47%)		200 (71.17%)	1 (0.36%)	
Olfactory loss	40 (14.23%)		241 (85.77%)	0 (0.00%)	
Gastrointestinal problems	70 (24.91%)		211 (75.09%)	0 (0.00%)	
**COVID-19 TESTING**					
**Attributes**	**Number of patients**				
	**Positive**		**Negative**	**Inconslusive or N/A**	
IgM (qualitatively)	225 (80.07%)		49 (17.44%)	7 (2.49%)	
IgG (qualitatively)	272 (96.80%)		2 (0.71%)	7 (2.49%)	
**VACCINATION**					
**Attributes**	**Number of patients**				
**Before 1st examination**	**Yes**		**No**		
1st dozen	11 (3.91%)		270 (96.09%)		
2nd dozen	3 (1.07%)		278 (1.07%)		
3rd dozen	1 (0.36%)		280 (99.64%)		
**Type**	**Corminaty**	**Spikevax**	**Vaxzevria**	**Janssen**	**N/A**
1st dozen	177 (62.99%)	19 (6.76%)	16 (5.69%)	16 (5.69%)	53 (18.87%)
2nd dozen	181 (64.41%)	17 (6.05%)	16 (5.69%)	0 (0.00%)	67 (23.85%)
3rd dozen	78 (27.76%)	0 (0.00%)	1 (0.36%)	12 (4.27%)	190 (67.61%)

**Table 2 diagnostics-13-01755-t002:** Performance comparison of selected machine learning algorithms.

Method	Accuracy	Balanced Accuracy	ROC-AUC	F1	Precision	Recall
**Logistic Regression**	61.40%	61.30%	63.33%	59.26%	59.26%	59.26%
**Multilayer Perceptron**	70.18%	69.81%	72.10%	66.67%	70.83%	62.96%
**Decision Tree**	**73.68%**	**73.52%**	**74.69%**	**71.70%**	**73.08%**	70.37%
**Random Forest**	70.18%	70.00%	70.62%	67.92%	69.23%	66.67%
***k*-Nearest Neighbors**	68.42%	68.15%	66.30%	65.38%	68.00%	62.96%
**Support Vector Machine**	63.16%	61.85%	70.37%	48.78%	71.43%	37.04%
**AdaBoost**	70.18%	70.37%	63.58%	70.18%	66.67%	**74.07%**
**XGBoost**	61.40%	61.11%	66.30%	57.69%	60.00%	55.56%
**Light Gradient Boosting Machine**	63.16%	62.59%	69.14%	57.14%	63.64%	51.85%

**Table 3 diagnostics-13-01755-t003:** Confusion matrix, Decision tree.

Actual label	Not recommended	23	7
	Recommended	8	19
		**Not recommended**	**Recommended**
		**Predicted label**	

## Data Availability

Dataset is submitted and will be published as a supplement to this paper in the journal Data. If you use Appendix A or data, please cite this paper.

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
