# Peer review of "Artificial-Intelligence-Driven Algorithms for Predicting Response to Corticosteroid Treatment in Patients with Post-Acute COVID-19"

_diagnostics, 2023, doi:10.3390/diagnostics13101755_

Round 1

Reviewer 1 Report

Overall, the manuscript report evaluation of the multiple AI models for the personalized selection of patients from the corticotherapy. However, there are still several places that need improve.

1. In the abstract, the authors should list the all AI model names they used in this study.

2. For the datasets, the authors mixed the patients screening and data recruitment for the AI models, which is a little difficult to understand clearly. The authors should separate them into two different sections. In particular, the authors should state clearly what the dataset includes for AI training.

3. Overall, all of those AI models is still low, when compared with other fields. Does the author analyze what the potential reasons for this? Or will they have some other factors affect this? If the author decrease some patient information into the AI training, will this increase the performance?

4. For Discussion, the authors may add more literature comparisons, and enhance the novelty and advantages when compare with the previous works.

Reviewer 2 Report

Dear Authors,

The article is very interesting, complex and innovating.

The era of COVID-19 pandemic, respectively, post-COVID-19 era represents one of the most important topics in medical domains thus we need such useful information as presented in this paper. The study is well designed, as well as literature data, and it brings value to our knowledge and understanding (and there is still much to find out in this specific area) since it is unique and it might represent a first step of a newly developed area of medical predictions and patients’ selection.

The approach based on artificial intelligence – driven algorithms has been used in several domains of medical and surgical specialties, but their use concerning the evolution of coronavirus infection is rather limited due to the novelty of the subject.

Here are my observations, suggestions:

1.    Abstract. The following statement “This paper also provides a simplified and easily human-interpretable Model” should be placed at Methods subsection before providing the results.

2.    Abstract. Conclusion. The final statement should be a clear take home message based on your original data.

3.    Keywords. I suggest adding “prediction model” in order to increase the readers’ interest.

4.    Line 51. The contributions of this paper are based on the actual research thus they should be placed at the end of the manuscript.

5.    It is not clear at all the structure of the paper. After the brief introduction, you should mention the aim then provide the methods and then the original results. The review should be placed after your original study or it may be a part of Discussion section.

6.    Figure 1. At the line of “start post-acute treatment” which is followed by “post-acute phase”: it seems (according to this representation) that the treatment starts before the actual diagnostic of this phase.

7.    Please use “COVID” with capital letters (for example, lines 319, 323).

8.    Limitations is part of Discussion section, not of results.

Well done!

Thank you

Round 2

Reviewer 1 Report

All of my concerns have been addressed. Thank you.